# Mixed methods protocol to examine the acceptability and clinical characteristics of a remote monitoring programme for delivery of COVID-19 care, among healthcare staff and patients

Robert Fox [1], Sophie Mulcahy Symmons,[1] Aoife De Brún,[1] David Joyce,[1] Eavan G Muldoon,[2,3] Tara McGinty,[2,3] Katherine M A O'Reilly,[3,4] Eileen O'Connor,[2] Eilish McAuliffe[1]

¹School of Nursing, Midwifery and Health Systems, University College Dublin, Dublin, Ireland
²Department of Infectious Diseases, Mater Misericordiae University Hospital, Dublin, Ireland
³School of Medicine, University College Dublin, Dublin, Ireland
⁴Department of Respiratory Medicine, Mater Misericordiae University Hospital, Dublin, Ireland

**Correspondence to**
Dr Robert Fox;
Robert.Fox@ucd.ie

## ABSTRACT

**Introduction** The use of remote monitoring technology to manage the care of patients with COVID-19 has been implemented to help reduce the burden placed on healthcare systems during the pandemic and protect the well-being of both staff and patients. Remote monitoring allows patients to record their signs and symptoms remotely (eg, while self-isolating at home) rather than requiring hospitalisation. Healthcare staff can, therefore, continually monitor their symptoms and be notified when the patient is showing signs of clinical deterioration. However, given the recency of the COVID-19 outbreak, there is a lack of research regarding the acceptance of remote monitoring interventions to manage COVID-19. This study will aim to evaluate the use of remote monitoring for managing COVID-19 cases from the perspective of both the patient and healthcare staff.

**Methods and analysis** Discharged patients from a large urban teaching hospital in Ireland, who have undergone remote monitoring for COVID-19, will be recruited to take part in a cross-sectional study consisting of a quantitative survey and a qualitative interview. A mixed methods design will be used to understand the experiences of remote monitoring from the perspective of the patient. Healthcare staff who have been involved in the provision of remote monitoring of patients with COVID-19 will be recruited to take part in a qualitative interview to understand their experiences with the process. Structural equation modelling will be used to examine the acceptance of the remote monitoring technology. Latent class analysis will be used to identify COVID-19 symptom profiles. Interview data will be examined using thematic analysis.

**Ethics and dissemination** Ethical approval has been granted by the ethical review boards at University College Dublin and the National Research Ethics Committee for COVID-19-related Research. Findings will be disseminated via publications in scientific journals, policy briefs, short reports and social media.

## Strengths and limitations of this study

► A mixed methods approach will allow for an in-depth examination of the remote monitoring programme for the management of patients with COVID-19.
► The perspective of both patients and healthcare staff will be examined, allowing for a more thorough evaluation of remote monitoring programmes for the management of COVID-19.
► The use of structural equation modelling techniques allows us to estimate more accurate parameter estimates in examining the user acceptability of remote monitoring programmes.
► As this study will use a sample of discharged patients with COVID-19 and healthcare staff from a hospital in Ireland, these findings may not be generalisable to patients and healthcare staff in other nations, or to patients who present with conditions other than COVID-19.
► This project will use a cross-sectional design; therefore, it will not be possible to infer the temporal ordering among the observed relationships.

## INTRODUCTION

SARS-CoV-2, which causes COVID-19, is a virus that, in the majority of clinical manifestations, resembles an influenza-like virus, including similar symptomatology such as cough, headache, fever and taste and smell disturbances.[1] However, patients with COVID-19 can exhibit heterogeneous symptom presentations with studies reporting varying symptom profiles in terms of symptom type (eg, respiratory, neurologic, gastrointestinal) and symptom severity, depending on factors such as age, sex, hospitalisation, comorbid health conditions and additional risk factors.[2–8] Moreover, COVID-19 appears to follow a biphasic pattern of illness consisting of an initial early viral response phase, followed by an exacerbated inflammatory second phase.[1 9] This second phase can be accompanied by respiratory compromise and hypoxia, including in cases whereby the initial phase was relatively

mild in terms of symptom presentation.[10] As such, it is important that patients with COVID-19 are continually monitored to ensure that appropriate interventions can take place in the event of clinical deterioration.

However, since the COVID-19 outbreak, healthcare systems have faced numerous challenges, including the loss of resource, mental and physical strain on patients and staff and staffing shortages due to high levels of COVID-19 among healthcare workers.[11–13] Moreover, individuals may delay seeking treatment[14] for other health conditions due to the fear and anxiety of contracting COVID-19, exacerbating health issues and increasing the subsequent burden on the healthcare system.[15] To effectively protect the well-being of patients and healthcare staff and to prevent the onward transmission of COVID-19, it is imperative that alternative initiatives are implemented to mitigate the impact of the COVID-19 pandemic on healthcare systems. These alternative initiatives may, therefore, allow the limited healthcare resources to be used most efficiently.

One such initiative is the remote monitoring of patients with COVID-19.[16–19] This is a system that involves the use of devices that allow healthcare staff at one location to monitor a patient at a different location. This allows for the home monitoring of patients, including physiological metrics such as oxygen saturation, body temperature and heart rate,[20] with mild to moderate symptoms (ie, who do not require hospitalisation) who are in self-isolation, tested positive for COVID-19 and/or are symptomatic. In addition, remote monitoring technology may be useful in monitoring postacute COVID-19 (or 'long-COVID'), whereby patients continue to exhibit some of the symptoms of COVID-19 despite having recovered.[21 22] Remote monitoring, as an alternative route for providing healthcare, has previously been found to be a useful and cost-effective means to managing patients across numerous types of conditions.[23–27] Recent research has demonstrated the feasibility of using remote monitoring technology to prospectively monitor respiratory illnesses such as common human coronaviruses, rhinovirus and respiratory syncytial virus.[28] However, given the recency of the COVID-19 pandemic, research pertaining to remote monitoring for managing patients with COVID-19 is limited. As such, it is important to examine different aspects of remote monitoring, such as patient and staff experiences and acceptability, to ensure its effectiveness as an alternative means to managing COVID-19 symptoms.

One important aspect to consider is the experience of the technology from the patient's perspective. Patient experience of the intervention is an important aspect in predicting treatment adherence.[29] One such means of modelling patient experience and adherence in remote monitoring-based interventions is through the technology acceptance model (TAM).[30] The TAM posits that the use of technology, such as remote monitoring, is the result of one's behavioural intention to use the technology. Behaviour intention, in turn, can be explained through one's perceived ease-of-use, perceived usefulness and attitude towards the technology. Alternative models of user acceptance (eg, the theory of planned behaviour (TPB), the unified theory of acceptance and use of technology (UTAUT) and the task-technology fit (TTF) models[31–33]) similarly posit that the use of the technology is the result of one's behavioural intention to use the technology. However, in predicting behavioural intention, these models also incorporate factors such as subjective norms/social influences that may play an important role in user acceptance.[31] These models aid in explaining user acceptance by providing a framework to incorporate psychosocial and behaviour theories such as the TPB[34] and the UTAUT[35] and have been applied to remote monitoring interventions across varying patient groups.[36–39]

Although research on remote monitoring for COVID-19 symptoms is scant, users of remote monitoring technologies for COVID-19 symptoms have generally reported high satisfaction and ease-of-use across interventions.[16 17] Moreover, findings suggest that patient's use of telemedicine systems can be effectively modelled within the TAM and TPB framework.[13 40 41] As such, these frameworks may be extended to modelling adherence in remote monitoring for patients with COVID-19.

Recent evidence, although limited, suggests healthcare staff generally have positive experiences with the use of telemedicine during the COVID-19 pandemic and that it may play an important role in their well-being.[42 43] For example, healthcare staff reported that they felt that the technology was easy to use and were satisfied with the safety precautions implemented to allow them to work from home.[42] Research indicates that healthcare staff generally hold positive views towards the remote monitoring of patients across numerous conditions.[44] However, there are notable concerns, such as managing increased workload and reduced quality of care from fewer patient visits.[44 45] This suggests that further research is required to better understand the staff experience of remote monitoring technology for patients with COVID-19, particularly across a diverse range of healthcare roles.

This project aims to evaluate a remote monitoring intervention for managing COVID-19 cases in a large urban teaching hospital in Dublin, Ireland. The objectives of this project are twofold. First, the acceptance of the remote monitoring technology from the perspective of the patient (objective 1a) and healthcare staff (objective 1b) will be examined using a mixed methods approach. User acceptance from the perspective of the patient will be examined using factors extracted from an integrated model of the UTAUT and TTF models,[33] while also controlling for a number of covariates, using quantitative analyses. User acceptance of remote monitoring for managing patients with COVID-19 from the perspective of both the patient and healthcare staff will also be examined using a qualitative methodology.

Second, patient symptom profiles of COVID-19 and postacute COVID-19 will be identified using a quantitative method approach (objective 2). Identifying patient symptom profiles of COVID-19 and postacute COVID-19

may highlight potential subgroups of patients who may be more likely to present with higher symptomatology or different sets of co-occurring symptoms. In addition, predictors, such as demographic variables and pre-existing comorbidities, of the COVID-19 symptom profiles will be determined. Identifying these predictors may be of particular benefit to identifying patients who may need to be monitored for a longer duration. For example, identifying patients who are likely to remain symptomatic (particularly symptoms such as shortness of breath) may need to be monitored for a longer duration after acute COVID-19. Furthermore, identifying such profiles may aid COVID-19-positive patients in being prepared for a likely set of symptoms that they may experience long term, depending on the predictors of the COVID-19 symptom profiles. For example, identifying patients who are likely to have persistent symptoms such as fatigue and loss of taste/smell may be useful in helping the patient being prepared for these long-term effects.

## METHODS
### Research design
A mixed methods design is proposed to achieve the research objectives of this project. The measures used to collect the data consist of both quantitative measures (eg, user acceptance of the remote monitoring technology) and qualitative interviews (for both patient and staff experience of remote monitoring). Work package 1 will use both quantitative cross-sectional survey data and qualitive interviews to examine patient experiences. Work package 1 will address objectives 1a and 2. Quantitative analyses will be used to determine the acceptance of the remote monitoring technology among the patients and to identify the patient symptom profiles of COVID-19 and postacute COVID-19. Work package 2 will use qualitative interviews to examine healthcare staff experiences. Work package 2 will address objective 1b.

### Participants and recruitment strategy
Participants will consist of patients and healthcare staff recruited from the clinic site. First, patients who have completed their treatment via remote monitoring will be recruited to gather information on their experiences of the intervention in the form of a quantitative survey. Patient recruitment will begin in February 2021 and end in June 2021. A total of 925 patients have been discharged from the remote monitoring service. Patients will be recruited via an email sent through gatekeepers at the clinic site. All patients will receive an email with the participant information sheet and a link to an online survey. A subsample of these patients will also be recruited for a qualitative interview regarding their experiences of remote monitoring. As part of the survey, patients have the option of indicating if they wish to participate in a telephone interview and provide their contact details. Patients who consented to interview will be purposively sampled to get a representative sample based on

age, gender and education. These patients will then be contacted by a member of the research team to arrange a suitable time for interview, once 7 days have passed since signing the consent form.

Second, staff who have been remotely monitoring patients with COVID-19 will be recruited via an email sent through gatekeepers at the clinic site. These participants will begin to be recruited in February 2021 and end in June 2021. A total of 25 staff worked in the virtual clinic since it was set up in April 2020. All staff who worked in the clinic will receive an invitation to participate in a qualitative interview regarding their experiences of remote monitoring from a healthcare staff perspective. If they choose to take part, a member of the research team will contact them to schedule a suitable time for a telephone interview, once 7 days have passed since signing the consent form.

The inclusion criteria for this project are that the patient or staff member must (a) be at least 18 years of age, (b) have the capacity to consent, (c) provide their full informed consent to take part in the study, (d) if they are a patient, they must have received a positive COVID-19 diagnosis, (e) if they are a patient, they must have undergone treatment via remote monitoring and (f) if they are a member of staff, they must have been monitoring the data and interacting with patients with COVID-19 for a minimum of 1 week. Patients who did not receive a positive diagnosis for COVID-19 and/or did not undergo treatment via remote monitoring will be excluded. The exclusion criterion for staff members is if they worked at the virtual clinic for less than 1 week.

### Measures: work package 1
#### Demographic information
Several demographics will be collected, including the patient's age (categorised as '18–29', '30–39', '40–49', 50–59', '60+'), sex, education (five categories ranging from 'primary education' to 'degree or postgraduate third-level education') and ethnicity.

#### COVID-19 symptomatology
Patient experience of COVID-19 will be measured via a 13-item measure that assesses the presence, or absence, of COVID-19 symptoms. This measure consists of 12 commonly reported COVID-19 symptoms (fever, cough, shortness of breath or difficulty with breathing, headaches, aches and pains, fatigue/tiredness, nausea, diarrhoea, sore throat, loss of appetite, loss of taste and loss of smell) and an additional open-ended option for 'other'. For each symptom, participants will be asked to indicate whether they experienced the symptom in the first 2 weeks following their diagnosis.

#### Postacute COVID-19 (long-COVID-19)
To assess for the presence, or absence, of postacute COVID-19 symptoms, patients will be presented with the same 13-item measure (fever, cough, shortness of breath or difficulty with breathing, headaches, aches and pains,

fatigue/tiredness, nausea, diarrhoea, sore throat, loss of appetite, loss of taste, loss of smell and an additional open-ended option for 'other') as the COVID-19 symptomatology questionnaire. However, they will be asked to indicate whether they continued experiencing the symptom for more than 2 weeks following their diagnosis and/or are still currently experiencing the symptom.

### Pre-existing comorbidities

Pre-existing comorbidities (ie, prior to a diagnosis of COVID-19) will be assessed via a nine-item list of physical comorbidities such as chronic respiratory disease and chronic heart disease. Responses will be scored using a trichotomous response format ('yes', 'no', 'unknown').

### Prior experience with mobile phone applications

To measure prior experience with smartphone applications, participants will be presented with five statements to which they chose the most appropriate statement (ranging from 'I have never used a mobile phone app prior to the COVID-19 app' to 'I use mobile phone apps regularly to upload and track my activity').

### User acceptance of remote monitoring equipment

Patients who report using either a pulse oximeter and a remote monitoring phone application, or just the remote monitoring phone application will be asked to complete a 16-item questionnaire pertaining to their experience of the technology. This multidimensional measure is comprised of items relating to the different factors of user technology acceptance, based on a recent integrated model consisting of factors from the UTAUT and TTF models[33]. Several factors will be extracted from this model, consisting of 'social influence' (two items; the degree to which important others agree to the use of the technology; for example, 'people who influence my behaviour thought it was important that I use the remote monitoring equipment'), 'facilitating conditions' (three items; individual's perception of the availability of resources to use the technology; for example, 'I felt I had the necessary knowledge to use the remote monitoring equipment'), 'effort expectancy' (four items; ease-of-use related to using the technology; for example, 'I find it easy to use such equipment'), 'performance expectancy' (four items; effectiveness to users in performing specific tasks; for example, 'I feel remote monitoring equipment is useful in obtaining health information') and 'TTF' (three items; the degree to which users believe that the performance of the technology matches its intended use; for example, 'In general, the remote monitoring equipment fully met my needs'). All items will be rated using a five-point Likert scale ('strongly disagree'=1, 'strongly agree'=5), with higher scores reflecting greater scores towards its respective factor. Previous psychometric research has supported the validity and reliability of these items.[33] Participants will also have the opportunity to share any additional information about their experience through an open-ended question.

### Adherence to the use of remote monitoring equipment

Adherence to the use of the pulse oximeter throughout the intervention will be assessed via three statements with participants being asked to choose the statement that best reflects their experience ('I did not use the oximeter', 'I tried to use the oximeter but gave up because it was too difficult', 'I occasionally used the oximeter during my treatment' and 'I used the oximeter consistently during my treatment'). In addition, participants will have the option to share any additional information about their experience of using the device through an open-ended question. Adherence to the use of the remote monitoring phone application will be assessed via three statements with participants being asked to choose the statement that best reflects their experience ('I did not use the phone application to record my symptoms', 'I tried to use the phone application but gave up because it was too difficult', 'I occasionally used the phone application to record my symptoms throughout my treatment' and 'I used the phone application to record my symptoms consistently throughout my treatment'). In addition, participants will have the option to share any additional information about their experience of using the phone application through an open-ended question.

### Perceived patient-centred care (patient–professional interaction)

Perceived patient-centred care will be assessed via the 16-item Patient-Professional Interaction Questionnaire (PPIQ).[46] The PPIQ examines patient-centred care by healthcare professional from the perspective of the patient by evaluating different aspects of the integration between the healthcare professional and the patient. This measure consists of four factors (each comprised of four items), which represents the care they received during the course of the remote monitoring intervention: effective communication (eg, 'he/she provided me with clear information'), interest in the patient's agenda (eg, 'he/she was interested in what I want from care'), empathy (eg, 'he/she understood my emotions') and patient involvement in care (eg, 'he/she gave me time to ask and to talk about the illness'. All items will be scored using a five-point Likert scale ('not at all'=1, 'very much'=5). Higher scores are indicative of better patient-centred care. The psychometric attributes of this measure have previously been supported.[46] Additionally, participants will have the option to provide any further information about their care through an open-ended question.

### Patient experience interview

Semistructured interviews will be conducted with patients who have undergone remote monitoring as part of their treatment for COVID-19 (n ≈ 20, or until saturation is reached). These interviews will be conducted by trained interviewers. A topic guide has been designed to explore the patient's experience of the remote monitoring intervention, including their: experience of having COVID-19; interactions with staff during the intervention; experience of the remote mentoring equipment; opinion on

remote monitoring compared with treatment-as-usual and opinions on using remote monitoring for other types of conditions. The interviews will take approximately 20 min. Interviews will be conducted via telephone on loudspeaker and recorded using audio-recorder or via video call on Zoom and the video call recorded to facilitate COVID-19 restrictions.

## Data analysis: work package 1

The first objective will be attained using a mixed methods design. First, descriptive statistics will assess the overall patient self-reported patient–professional interaction, adherence to the use of the remote monitoring equipment and acceptance towards the remote monitoring technology and provide the sample characteristics.

Second, structural equation modelling will be used to examine different aspects of user acceptance towards remote monitoring technology, such as the performance expectancy of the equipment from the perspective of the patient. Structural equation modelling is advantageous as it can parse out measurement error, thereby yielding more accurate parameter estimates[47] and can be applied to medical research.[48 49] We will also determine the effect of several exogenous covariates such as demographics, pre-existing comorbidities, prior experience with mobile phone applications and patient–professional interaction in predicting user acceptance towards the use of remote monitoring technology. Given the larger sample size required for structural equation modelling, in the event that an insufficient sample size is gathered, then a multiple regression analysis will be conducted instead.

Third, latent classes, or symptom profiles, of COVID-19 and postacute COVID-19 will be examined through latent class analysis (LCA), using robust maximum likelihood estimation. To determine the optimal number of latent classes, models with one to six classes will be examined. To avoid solutions based on local maxima, 500 random sets of starting values will be used followed by 100 final stage optimisations. Several fit indices will be used to determine the fit of each latent class model, including the Akaike information criterion (AIC),[50] the Bayesian information criterion (BIC),[51] the sample size-adjusted BIC (ssaBIC),[52] entropy values and the Lo-Mendell-Rubin-adjusted likelihood ratio test (LMR-A).[53] Lower AIC, BIC and ssaBIC values and higher entropy values are indicative of better model fit. A non-significant LMR-A value suggests that the model with one less class should be accepted. In the event that an insufficient sample size is gathered, then a cluster analysis will be conducted instead.

Next, to determine the predictors of each symptom profile, a multinomial logistic regression will be performed by regressing the latent classes (identified during the class enumeration process) onto several covariates, using the R3STEP function in Mplus.[54 55] This three-step procedure involves first identifying the most appropriate latent classes; then obtaining the most likely class memberships based on the posterior probabilities of the LCA, while accounting for the classification uncertainty rate (ie, measurement error) and, finally, the most likely class memberships are analysed with the covariates, thereby accounting for at least some of the misclassification errors.[55 56]

Fourth, the interviews will be transcribed verbatim. A thematic analysis will be conducted using NVivo V.12[57] to identify common themes throughout the interviews. An inductive approach will be taken to draw out themes, without a pre-existing theory in the literature or based on the researchers' preconceptions, by one researcher through familiarisation with the data, initial coding, refinement and subsequently grouped into themes to best represent the data. It will then be reviewed by the research team to ensure that all emerging topics are included and consensus is reached. A secondary researcher will independently code a subset of transcripts to assess the internal reliability.

## Measures: work package 2
### Staff experience interview

Semistructured interviews will be conducted with up to 15 (or until saturation in the data is reached) healthcare staff who were responsible for monitoring the data and interacting with patients with COVID-19 for a minimum of at least 1 week at the clinic and agree to take part in the study. These interviews will be conducted by trained interviewers and will take approximately 30 min. Interviews will be conducted over the phone and recorded using audio-recorder or via video call on Zoom and the video call recorded. A topic guide was developed to explore the staff member's experience of the remote monitoring process, including their role in the clinic; the usefulness of equipment such as pulse oximeters; how well the equipment was received by patients; their experiences of the remote monitoring intervention compared with treatment-as-usual, such as interactions with patients; the impact of remote monitoring on reducing the burden placed on healthcare staff during the pandemic; the benefits and drawback of using remote monitoring; how remote monitoring can be improved and their opinions on using remote monitoring for other types of conditions.

## Data analysis: work package 2

To achieve the second objective of this project, the data collected during the qualitative interviews with the healthcare staff members will be analysed. These interviews will be transcribed verbatim, and a thematic analysis will be conducted using NVivo to identify common themes throughout the interviews. An inductive approach will be taken to draw out themes by one researcher and will then be reviewed by the research team to ensure that all emerging topics are included, following the same steps as work package 1. A secondary researcher will independently code a subset of transcripts to assess the internal reliability.

## Patient and public involvement

Patients and/or the public were not involved in the design of this study protocol.

## Ethics and dissemination

Ethical approval has been granted by the National Research Ethics Committee for COVID-19-related Research (NREC COVID-19; reference number: 20-NREC-COV-093). Data sharing agreements have been put in place between the University College Dublin (UCD) research team and the hospital. The data will be deidentified and securely transferred to the research team, in accordance with data protection regulations. No identifiable data will be included in the data set received by the research team at UCD. Online consent will be obtained from the participants of this project.

Findings will be disseminated via publications in scientific journals, policy briefs, short reports and social media. A summary of the findings will also be shared with participants who informed the research team that they are interested in the results of the project.

**Contributors** RF, SMS, ADB, DJ and EMcA contributed to the conceptualisation, planning and design of the project; EMcA obtained funding and ethical approval for the project; RF, SMS, ADB, DJ and EMcA drafted the initial protocol; EGM, TM, KOR and EOC provided critical input into the protocol; RF, ADB, SMS, DJ, EGM, TM, KOR, EOC and EMcA contributed to the writing, revising and editing of the protocol. All authors contributed to and have approved the final protocol.

**Funding** This work was supported by the Science Foundation Ireland (SFI). Grant number: 20/COV/0221.

**Competing interests** None declared.

**Patient and public involvement** Patients and/or the public were not involved in the design, or conduct, or reporting, or dissemination plans of this research.

**Patient consent for publication** Not applicable.

**Provenance and peer review** Not commissioned; externally peer reviewed.

**ORCID iD**
Robert Fox http://orcid.org/0000-0002-0950-3865

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
