## [Reviewer comments · BMJ Open]

ARTICLE DETAILS

TITLE (PROVISIONAL)	A mixed methods protocol to examine the acceptability and clinical characteristics of a remote monitoring programme for delivery of COVID-19 care, among healthcare staff and patients.
AUTHORS	Fox, Robert; Symmons, Sophie Mulcahy; De Brún, Aoife; Joyce, David; Muldoon, Eavan G; McGinty, Tara; O'Reilly, Katherine M.A.; O'Connor, Eileen; McAuliffe, Eilish

VERSION 1 – REVIEW

REVIEWER	Akbarialiabad, Hossein Shiraz University of Medical Sciences
REVIEW RETURNED	12-Apr-2021

GENERAL COMMENTS	I read your proposed protocol very carefully. This is a very practical and time-sensitive issue, so I personally prioritized reviewing this paper to accelerate the process. The importance of the work is well noted. Abstract and introduction were written very well. I have a few points in methods : 1-I have one point that is recommended to be addressed by authors. The positive perception of the remote care of COVID-19 patients by the staff and patients does not necessarily correlate with improved patient outcomes. I assume it would be more meaningful if the authors could compare the impact between those who underwent this management strategy vs. those who were followed by routine followups, for example, in clinics. This is not an obligation, in my view, just to be considered by authors. 2- in the research design of the method (page 10), quantitative phase of the study is not clear. You may add a few sentences in the research design for better clarity. 3- Did you evaluate the reliability and validity of the questionnaire? This should be evaluated with a pilot sample in my view. 4- In the inclusion criteria, the authors mentioned : if they are a member of staff, they must have been monitoring the data and interacting with COVID-19 patients for a minimum of one week.
--

	I think the threshold should be higher for the staff. I assume the time required for a proper reflection would be more than this. For example, an experience of at least two or three weeks is more meaningful to me. 5- On the section of Post-acute COVID-19 (long-COVID) : I suggest looking into papers on long COVID. I have one review paper in mind and I suggest looking into such papers. https://www.preprints.org/manuscript/202103.0490/v1 I suggest using PASC (Post-Acute Sequelae of COVID-19) instead (or as well as) of using post-acute COVID as it is gaining more popularity in this literature for your prospective citation of the work. There are other questions that the authors could ask in their cross-sectional study for example hair loss and other signs/symptoms easy to detect in long COVID. Go to preprint: section of sign/symptoms and especially "Multisystem studies and those related to general signs and symptoms" Pain/aches are very common words. It may better to mention the relevant examples such as joint pain (arthralgia) and chest pain, etc. Timing for long COVID research is usually three weeks and beyond following the diagnosis. (nomenclature section) 6- In "Work package 2, Staff experience interview", the researchers mentioned that they will interview 10 staff. how can you be sure that 10 healthcare staff would be sufficient to reach saturation? I hope these comments be helpful to the authors So I suggest publication with minor revision.
--	---

REVIEWER	Silva, Cícera Federal University of Campina Grande Centre of Teacher Education
REVIEW RETURNED	19-Apr-2021

GENERAL COMMENTS	The study is very relevant, in view of the epidemiological moment that we are living, of the pandemic of COVID-19, and the growth of remote monitoring of patients. There are some weaknesses in this protocol, which should be noted by the authors, and I have a few questions, which I briefly describe below. 1) It would be interesting if the title had the identification that it is a protocol. 2) In the Abstract there is no brief description of the analysis of the quantitative and qualitative data.
--

	3) The stated objectives are less comprehensive than the objectives that we can perceive in the description of the study. 4) In Methods (Research design), there is talk of quantitative measures (reflecting patient experience ...). The reflection on the lived experience is an objective of the qualitative methodology. 5) In the description of the participants, it is necessary to communicate data on the number of health professionals working in the service, as well as the number of patients seen so far, to give an idea of the study population. It talks about the approximate number of people (patients and professionals) who will be interviewed, but we have no idea about the study population. 6) The inclusion criteria are insufficient and there are no exclusion criteria reported. 7) The participants' sampling procedure for the qualitative stage is not described. Will it be theoretical saturation sampling? It is important to provide more details. 8) Was the data collection instruments validated? It is recommended to offer this information. 9) The authors address the transcription of the interviews, but do not mention the procedures for recording the interviewees' voice. 10) It is necessary to mention which theoretical framework will be used in the analysis of qualitative data. 11) What is the relevance of comparing interviews between groups (age, sex) for this study? It must be described. 12) Regarding ethical aspects, it is important to include the number of the approval opinion by the Ethics Committee. 13) The authors bring the STROBE checklist. But the COREQ checklist should also be mentioned, for the qualitative part of the study. In addition, it is interesting to report the planning of the use of the specific checklist for assessing the quality of studies of the mixed methods type (Mixed Methods Appraisal Tool). 14) What is the current study schedule? For it is reported that the recruitment date was February 2021.
--	---

REVIEWER	Rod, J. Queensland University of Technology
REVIEW RETURNED	01-May-2021

GENERAL COMMENTS	Thanks for the opportunity of reviewing this protocol. I think the research questions and aims are clear. Nevertheless, I found it difficult to understand the rationale behind exploring patient symptoms profiles, I could not find an explanation of the need for this analysis within the protocol. The theoretical background seems to be solid. However, I think there is room for more linking between the theory and the measures or constructs of the theory (see comments below). Overall, I think the research proposal is of value and importance, the proposed methodology of SEM and thematic analysis also seem appropriate for the research question. I would just suggest to the authors to have a look at the required sample size for conducting SEM based on their hypothesized effect estimate results. There might be a methodological need for other types of analysis to explore the desired relationships. Perhaps it would be useful to know if there is a plan b for the analysis? 1- Abstract section
--

	Lines 19-24: there is a lack of research regarding the effectiveness of remote monitoring in covid-19. Thus, the study will evaluate the use of remote monitoring by patients and staff. The study will focus on understanding subjective experiences of patients and staff. Although subjective experiences might give some support to the usefulness or value in terms of subjective experience or perceived quality of the service, it does not produce an overall conclusion about the effectiveness (in the general clinical or economic sense). Usually, evaluation of effectiveness rely on concrete endpoints that are used to measure clinical or and economic improvement. I would recommend rephrasing the first part of the sentence given that it might be misleading. Or perhaps, it would be worthwhile to consider if acceptance of the technology is a “first step?”, or part of the process of evaluating effectiveness. In any case, I think that it should be clear that acceptance does not immediately imply effectiveness. 2- Strengths and limitations section Similar to the above comment, it is not clear what is meant by “effective evaluation”, the methods seem to indicate the examination of user acceptability and not an undefined “effectiveness” of remote monitoring. 3- Introduction section Is SARS-COV2 still a novel virus?. I am not sure. Aims: There seem to be two aims based on the methods sections.  • To evaluate the use of remote monitoring from the (a) patients and (b) staff perspective • To identify patient symptom profiles of COVID-19 and post COVID-19. The rationale for conducting the first aim is explained in the introduction. Here I would just recommend paying attention to the terminology as stated above. However, the introduction does not explain what is the value of studying the predictors of particular symptoms profiles. I am not proposing that there is no value in pursuing this aim, but the authors should present it explicitly. Perhaps I did not understand it?. I think an effort should be made to make the value of this aim clearer. Additionally, at present, I also find it hard to see the relationship between the predictors for symptomatology and the evaluation of patient experiences. I would suggest considering writing the aims as above and them explaining in the methods that aim 1a and 2 is addressed by work package 1 and that aim 1b is addressed by work package 2. Alternatively, perhaps drop aim number 2 and just focus on patient experiences unless there is a clear link between the two aims? 4- Methods section
--	---

	Work package 1: a. Measures sub-section: I think the sub-section quality of care should be named "perceived quality of care" as it is a self-reported measure. Lines 42-45: I could not find the items measuring the "perceived usefulness" Line 56: I could not find the items measuring "behavioral intentions" How were the questions of "perceived usefulness" and "behavioral intentions" formulated? b. Patient experience of remote monitoring equipment sub-section The TAM and the TBP were mentioned in the introduction, nevertheless is not clear whether questions regarding user acceptance were formulated based on these theoretical models?. Which model will be used in the analysis? It is a combination? Should this section be named user acceptance?
--	---

VERSION 1 – AUTHOR RESPONSE

Reviewers' comments:

Reviewer #1:

Thank you for submitting your protocol to BMJ open

I read your proposed protocol very carefully. This is a very practical and time-sensitive issue, so I personally prioritized reviewing this paper to accelerate the process.

The importance of the work is well noted. Abstract and introduction were written very well.

Response: Thank you for the kind words, they are greatly appreciated.

1-I have one point that is recommended to be addressed by authors. The positive perception of the remote care of COVID-19 patients by the staff and patients does not necessarily correlate with improved patient outcomes. I assume it would be more meaningful if the authors could compare the impact between those who underwent this management strategy vs. those who were followed by routine followups, for example, in clinics. This is not an obligation, in my view, just to be considered by authors.

Response: Thank you for this suggestion. We agree that this would be a useful addition to our study. However, unfortunately, we do not have access to such data to include as a comparison.

2- in the research design of the method (page 10), quantitative phase of the study is not clear. You may add a few sentences in the research design for better clarity.

Response: We have now included additional information to clarify this.

Methods (pg 8):

“Work Package 1 will use both quantitative cross-sectional survey data and qualitative interviews to examine patient experiences. Work Package 1 will address objectives 1a and 2. Quantitative analyses will be used to determine the acceptance of the remote monitoring technology among the patients, and to identify the patient symptom profiles of COVID-19 and post-acute COVID-19. Work Package 2 will use qualitative interviews to examine healthcare staff experiences. Work package 2 will address objective 1b.”

3- Did you evaluate the reliability and validity of the questionnaire? This should be evaluated with a pilot sample in my view.

Response: The reliability and validity of the quantitative questionnaires used to assess user acceptance of the remote monitoring technology and the patient-professional interaction have previously been supported. References to these studies are now included in the methods section. As the data collection has already commenced and the psychometric attributes of some of the questionnaires have been previously supported, we do not believe that carrying out a pilot will be necessary.

Research demonstrating the reliability and validity of the user acceptance models used (pg 11):
“Previous psychometric research has supported the validity and reliability of these items.[33]”

Research demonstrating the reliability and validity of the patient-professional interaction questionnaire used (pg 12):
“The psychometric attributes of this measure have previously been supported.[46]”

4- In the inclusion criteria, the authors mentioned:

If they are a member of staff, they must have been monitoring the data and interacting with COVID-19 patients for a minimum of one week.

I think the threshold should be higher for the staff. I assume the time required for a proper reflection would be more than this. For example, an experience of at least two or three weeks is more meaningful to me.

Response: Thank you for this suggestion. Although we believe that one week will be sufficient time to reflect on the healthcare staff's experience, our experience of data collection thus far has indicated that all respondents have worked greater than one week, generally at least one month.

5- On the section of Post-acute COVID-19 (long-COVID) :

I suggest looking into papers on long COVID. I have one review paper in mind and I suggest looking into such papers.

<https://www.preprints.org/manuscript/202103.0490/v1>

I suggest using PASC (Post-Acute Sequelae of COVID-19) instead (or as well as) of using post-acute COVID as it is gaining more popularity in this literature for your prospective citation of the work. There are other questions that the authors could ask in their cross-sectional study for example hair loss and other signs/symptoms easy to detect in long COVID. Go to preprint: section of sign/symptoms and especially "Multisystem studies and those related to general signs and symptoms"

Pain/aches are very common words. It may better to mention the relevant examples such as joint pain (arthralgia) and chest pain, etc.

Response: Thank you for sharing this research. Unfortunately, given our current stage of data collection, it would not be possible to change our instruments used. However, we believe that the current set of symptoms assessed, alongside an open-ended option for the participant to include any additional symptoms, will be sufficient to obtain our objectives of this research.

Timing for long COVID research is usually three weeks and beyond following the diagnosis.
(nomenclature section)

Response: Unfortunately, as the data collection commenced in February, we cannot change the two-week marker for COVID-19 symptoms. However, as we have assessed symptoms of COVID-19 that are still being experienced by the patient, we can identify symptoms that are still occurring to date (i.e., those beyond three weeks, depending on the date of diagnosis). Moreover, this allows us to remain with a clear timeline examining the course of COVID-19 symptoms (initial symptoms, those at two weeks, and those still being experienced).

6- In "Work package 2, Staff experience interview", the researchers mentioned that they will interview 10 staff.

How can you be sure that 10 healthcare staff would be sufficient to reach saturation?

Response: We have now increased our aim for the number of staff to 15 participants. However, as we have a small sample to select from (a total of 25 staff members), we may not achieve this goal.

Methods (pg 15):

"Semi-structured interviews will be conducted with up to 15 (or until saturation in the data is reached) healthcare staff who were responsible for monitoring the data and interacting with COVID-19 patients for a minimum of at least one week at the clinic and agree to take part in the study."

We have also included information pertaining to the total number of staff who worked in the virtual clinic in the Methods section (pg 9):

"A total of 25 staff worked in the virtual clinic since it was set up in April 2020"

We would like to thank you for taking the time to review our manuscript and for providing feedback that allows us to improve the quality of our work.

Reviewer #2:

The study is very relevant, in view of the epidemiological moment that we are living, of the pandemic of COVID-19, and the growth of remote monitoring of patients.

Response: Thank you for the kind words.

There are some weaknesses in this protocol, which should be noted by the authors, and I have a few questions, which I briefly describe below.

1) It would be interesting if the title had the identification that it is a protocol.

Response: We have now changed the title of the protocol as follows:

“A mixed methods protocol to examine the acceptability and clinical characteristics of a remote monitoring programme for delivery of COVID-19 care, among healthcare staff and patients.”

2) In the Abstract there is no brief description of the analysis of the quantitative and qualitative data.

Response: Thank you for noting this oversight. We have now included the information in the abstract (pg 2):

“Structural equation modelling will be used to examine the acceptance of the remote monitoring technology. Latent class analysis will be used to identify COVID-19 symptom profiles. Interview data will be examined using thematic analysis”

3) The stated objectives are less comprehensive than the objectives that we can perceive in the description of the study.

Response: We have now included additional information when detailing the objectives of the protocol.

Introduction (pg 7):

“First, the acceptance of the remote monitoring technology from the perspective of the patient (objective 1a) and healthcare staff (objective 1b) will be examined using a mixed methods approach. User acceptance from the perspective of the patient will be examined using factors extracted from an integrated model of the UTAUT and TTF models (see [33]), while also controlling for a number of covariates, using quantitative analyses. User acceptance of remote monitoring for managing COVID-19 patients from the perspective of both the patient and healthcare staff will also be examined using a qualitative methodology.

Second, patient symptom profiles of COVID-19 and post-acute COVID-19 will be identified using a quantitative methods approach (objective 2). Identifying patient symptom profiles of COVID-19 and post-acute COVID-19 may highlight potential subgroups of patients who may be more likely to present with higher symptomatology or different sets of co-occurring symptoms. In addition, predictors, such as demographic variables and pre-existing comorbidities, of the COVID-19 symptom profiles will be determined. Identifying these predictors may be of particular benefit to identifying patients who may need to be monitored for a longer duration. For example, identifying patients who are likely to remain symptomatic (particularly symptoms such as shortness of breath) may need to be monitored for a longer duration after acute COVID-19. Furthermore, identifying such profiles may aid COVID-19 positive patients in being prepared for a likely set of symptoms that they may experience long-term, depending on the predictors of the COVID-19 symptom profiles. For example, identifying patients who are likely to have persistent symptoms such as fatigue and loss of taste/smell may be useful in helping the patient being prepared for these long-term effects.”

4) In Methods (Research design), there is talk of quantitative measures (reflecting patient experience ...). The reflection on the lived experience is an objective of the qualitative methodology.

Response: This section has now been rephrased for clarity:

Methods (pg 8):

“The measures used to collect the data consists of both quantitative measures (e.g., user acceptance of the remote monitoring technology) and qualitative interviews (for both patient and staff experience of remote monitoring).”

5) In the description of the participants, it is necessary to communicate data on the number of health professionals working in the service, as well as the number of patients seen so far, to give an idea of the study population. It talks about the approximate number of people (patients and professionals) who will be interviewed, but we have no idea about the study population.

Response: Thank you for noting this. We have now included this information in the Methods section to reflect this.

Methods (pg 8):

“A total of 925 patients have been discharged from the remote monitoring service”

Methods (pg 9):

“A total of 25 staff worked in the virtual clinic since it was set up in April 2020”

6) The inclusion criteria are insufficient and there are no exclusion criteria reported.

Response: We have included additional information regarding the inclusion and exclusion criteria.

Methods (pg 9):

“The inclusion criteria for this project are that the patient or staff member must (a) be at least 18 years of age, (b) have the capacity to consent, (c) provide their full informed consent to take part in the study, (d) if they are a patient, they must have received a positive COVID-19 diagnosis, (e) if they are a patient, they must have undergone treatment via remote monitoring, and (f), if they are a member of staff, they must have been monitoring the data and interacting with COVID-19 patients for a minimum of one week. Patients who did not receive a positive diagnosis for COVID-19 and/or did not undergo treatment via remote monitoring will be excluded. The exclusion criterion for staff members is if they worked at the virtual clinic for less than one week.”

7) The participants' sampling procedure for the qualitative stage is not described. Will it be theoretical saturation sampling? It is important to provide more details.

Response: A purposive sampling approach will be taken to gain a representative sample across age, gender, and education. We have now included this information.

Methods (pg 8):

“Patients who consented to interview will be purposively sampled to get a representative sample based on age, gender, and education.”

8) Was the data collection instruments validated? It is recommended to offer this information.

Response: The user acceptance and patient-professional interaction questionnaires have previously been validated. This information is now included in the Methods section.

Research demonstrating the reliability and validity of the user acceptance models used (pg 11):

“Previous psychometric research has supported the validity and reliability of these items.[33]”

Research demonstrating the reliability and validity of the patient-professional interaction questionnaire used (pg 12):

“The psychometric attributes of this measure have previously been supported.[46]”

9) The authors address the transcription of the interviews, but do not mention the procedures for recording the interviewees' voice.

Response: The interviews will be conducted and audio-recorded via Zoom or via telephone and recorded using an audio-recorder. The Methods section has been updated to include this information.

Patient interviews Methods (pg 13):

“The interviews will take approximately 20 minutes. Interviews will be conducted via telephone on loudspeaker and recorded using audio-recorder or via video call on Zoom and the video call recorded to facilitate COVID-19 restrictions.”

Staff interviews Methods (pg 15):

“These interviews will be conducted by trained interviewers and will take approximately 30 minutes. Interviews will be conducted over the phone and recorded using audio-recorder or via video call on Zoom and the video call recorded.”

10) It is necessary to mention which theoretical framework will be used in the analysis of qualitative data.

Response: As we are using an inductive approach, we do not believe that we should approach the qualitative data analysis with an existing theoretical framework. Moreover, we plan on using thematic analysis to analyse the data. This is a widely used, flexible approach to qualitative analysis, which is not based in pre-existing theories, that can help interpret patterns of meaning in the data. This is particularly true of inductive data (i.e., not having pre-existing expectations of the data). We have now noted this absence of a theoretical framework in detailing our analytical plan.

Methods (pg 15):

“An inductive approach will be taken to draw out themes, without a pre-existing theory in the literature or based on the researchers' preconceptions, by one researcher through familiarisation with the data, initial coding, refinement and subsequently grouped into themes to best represent the data.”

11) What is the relevance of comparing interviews between groups (age, sex) for this study? It must be described.

Response: Thank you for highlighting this. On reflection, we have decided to remove these comparisons from our analytical plan.

12) Regarding ethical aspects, it is important to include the number of the approval opinion by the Ethics Committee.

Response: This information has now been included in the Ethics and Dissemination section (pg 16):

“Ethical approval has been granted by the National Research Ethics Committee for COVID-19-related Research (NREC COVID-19; reference number: 20-NREC-COV-093).”

13) The authors bring the STROBE checklist. But the COREQ checklist should also be mentioned, for the qualitative part of the study. In addition, it is interesting to report the planning of the use of the specific checklist for assessing the quality of studies of the mixed methods type (Mixed Methods Appraisal Tool).

Response: We have now included the COREQ checklist as supplemental material to the protocol.

14) What is the current study schedule? For it is reported that the recruitment date was February 2021.

Response: The data collection commenced in February 2021 and will end in June 2021. The Methods section has been updated to reflect this.

Thank you for taking the time to review our manuscript and for providing such helpful feedback.

Reviewer #3:

I think the research questions and aims are clear. Nevertheless, I found it difficult to understand the rationale behind exploring patient symptoms profiles, I could not find an explanation of the need for this analysis within the protocol.

Response: Thank you for the kind words. We have now included a rationale for exploring patient symptom profiles.

Introduction (pg 7):

“Second, patient symptom profiles of COVID-19 and post-acute COVID-19 will be identified using a quantitative methods approach (objective 2). Identifying patient symptom profiles of COVID-19 and post-acute COVID-19 may highlight potential subgroups of patients who may be more likely to present with higher symptomatology or different sets of co-occurring symptoms. In addition, predictors, such as demographic variables and pre-existing comorbidities, of the COVID-19 symptom profiles will be determined. Identifying these predictors may be of particular benefit to identifying patients who may need to be monitored for a longer duration. For example, identifying patients who are likely to remain symptomatic (particularly symptoms such as shortness of breath) may need to be monitored for a longer duration after acute COVID-19. Furthermore, identifying such profiles may aid COVID-19 positive patients in being prepared for a likely set of symptoms that they may experience long-term, depending on the predictors of the COVID-19 symptom profiles. For example, identifying patients who are likely to have persistent symptoms such as fatigue and loss of taste/smell may be useful in helping the patient being prepared for these long-term effects.”

The theoretical background seems to be solid. However, I think there is room for more linking between the theory and the measures or constructs of the theory (see comments below).

Overall, I think the research proposal is of value and importance, the proposed methodology of SEM and thematic analysis also seem appropriate for the research question. I would just suggest to the

authors to have a look at the required sample size for conducting SEM based on their hypothesized effect estimate results. There might be a methodological need for other types of analysis to explore the desired relationships. Perhaps it would be useful to know if there is a plan b for the analysis?

Response: We have now included a backup plan for the SEM analysis and the latent class analysis.

For the SEM analysis in the Methods section (pg 14):

“Given the larger sample size required for structure equation modelling, in the event that an insufficient sample size is gathered, then a multiple regression analyse will be conducted instead.”

For the latent class analysis in the Methods section (pg 14):

“In the event that an insufficient sample size is gathered, then a cluster analysis will be conducted instead.”

1- Abstract section

Lines 19-24: there is a lack of research regarding the effectiveness of remote monitoring in covid-19. Thus, the study will evaluate the use of remote monitoring by patients and staff.

The study will focus on understanding subjective experiences of patients and staff. Although subjective experiences might give some support to the usefulness or value in terms of subjective experience or perceived quality of the service, it does not produce an overall conclusion about the effectiveness (in the general clinical or economic sense). Usually, evaluation of effectiveness rely on concrete endpoints that are used to measure clinical or and economic improvement.

I would recommend rephrasing the first part of the sentence given that it might be misleading. Or perhaps, it would be worthwhile to consider if acceptance of the technology is a “first step?”, or part of the process of evaluating effectiveness. In any case, I think that it should be clear that acceptance does not immediately imply effectiveness.

Response: Thank you for noting this. We agree with this point. We have rephrased the sentence to reflect “acceptance” rather than “effectiveness” of remote monitoring.

Abstract (pg 2):

“However, given the recency of the COVID-19 outbreak, there is a lack of research regarding the acceptance of remote monitoring interventions to manage COVID-19.”

2- Strengths and limitations section

Similar to the above comment, it is not clear what is meant by “effective evaluation”, the methods seem to indicate the examination of user acceptability and not an undefined “effectiveness” of remote monitoring.

Response: Our point was made in reference to the advantage of a mixed methods approach being advantageous as it provides an in-depth examination, compared to using either a solely quantitative or qualitative approach. This has now been rephased for clarity. We have now rephrased this to “in-depth examination” rather than “effective evaluation”.

Strengths and limitations (pg 3):

“A mixed methods approach will allow for an in-depth examination of the remote monitoring programme for the management of COVID-19 patients.”

3- Introduction section

Is SARS-COV2 still a novel virus?. I am not sure.

Response: We have removed the implication that SARS-CoV-2 is a novel virus.

Aims: There seem to be two aims based on the methods sections.

- To evaluate the use of remote monitoring from the (a) patients and (b) staff perspective
- To identify patient symptom profiles of COVID-19 and post COVID-19.

The rationale for conducting the first aim is explained in the introduction. Here I would just recommend paying attention to the terminology as stated above. However, the introduction does not explain what is the value of studying the predictors of particular symptoms profiles.

I am not proposing that there is no value in pursuing this aim, but the authors should present it explicitly. Perhaps I did not understand it?. I think an effort should be made to make the value of this aim clearer. Additionally, at present, I also find it hard to see the relationship between the predictors for symptomatology and the evaluation of patient experiences.

I would suggest considering writing the aims as above and then explaining in the methods that aim 1a and 2 is addressed by work package 1 and that aim 1b is addressed by work package 2.

Alternatively, perhaps drop aim number 2 and just focus on patient experiences unless there is a clear link between the two aims?

Response: Thank you for this suggestion. We agree that restating the objectives as 1a and 1b reflecting the patient and staff experiences, respectively, and objective 2 as reflecting the COVID-19 symptom profiles is a clearer way to present our study. We have now included this in-text. Moreover, we have included a rationale for examining the patient symptom profiles of COVID-19.

Introduction (pg 7):

“First, the acceptance of the remote monitoring technology from the perspective of the patient (objective 1a) and healthcare staff (objective 1b) will be examined using a mixed methods approach. User acceptance from the perspective of the patient will be examined using factors extracted from an integrated model of the UTAUT and TTF models (see [33]), while also controlling for a number of covariates, using quantitative analyses. User acceptance of remote monitoring for managing COVID-19 patients from the perspective of both the patient and healthcare staff will also be examined using a qualitative methodology.

Second, patient symptom profiles of COVID-19 and post-acute COVID-19 will be identified using a quantitative methods approach (objective 2). Identifying patient symptom profiles of COVID-19 and post-acute COVID-19 may highlight potential subgroups of patients who may be more likely to present with higher symptomatology or different sets of co-occurring symptoms. In addition, predictors, such as demographic variables and pre-existing comorbidities, of the COVID-19 symptom profiles will be determined. Identifying these predictors may be of particular benefit to identifying patients who may need to be monitored for a longer duration. For example, identifying patients who are likely to remain symptomatic (particularly symptoms such as shortness of breath) may need to be monitored for a longer duration after acute COVID-19. Furthermore, identifying such profiles may aid COVID-19

positive patients in being prepared for a likely set of symptoms that they may experience long-term, depending on the predictors of the COVID-19 symptom profiles. For example, identifying patients who are likely to have persistent symptoms such as fatigue and loss of taste/smell may be useful in helping the patient being prepared for these long-term effects.”

Methods (pg 8):

“Work Package 1 will use both quantitative cross-sectional survey data and qualitative interviews to examine patient experiences. Work Package 1 will address objectives 1a and 2. Quantitative analyses will be used to determine the acceptance of the remote monitoring technology among the patients, and to identify the patient symptom profiles of COVID-19 and post-acute COVID-19. Work Package 2 will use qualitative interviews to examine healthcare staff experiences. Work package 2 will address objective 1b.”

4- Methods section

Work package 1:

a. Measures sub-section:

I think the sub-section quality of care should be named "perceived quality of care" as it is a self-reported measure.

Response: We have renamed to section to “perceived patient-centred care (patient-professional interaction)” to reflect the self-report nature of the questionnaire.

Lines 42-45: I could not find the items measuring the “perceived usefulness”

Line 56: I could not find the items measuring “behavioral intentions”

How were the questions of “perceived usefulness” and “behavioral intentions” formulated?

Response: The items were formulated using an integrated model of the UTAUT and TTF models of user acceptance. Further details regarding this questionnaire are now included, such as sample items and support for psychometric attributes.

Methods (pg 11):

“This multidimensional measure is comprised of items relating to the different factors of user technology acceptance, based on a recent integrated model consisting of factors from the UTAUT and TTF models (see [33]). Several factors will be extracted from this model, consisting of ‘social influence’ (two items; the degree to which important others agree to the use of the technology; e.g., “people who influence my behaviour thought it was important that I use the remote monitoring equipment”), ‘facilitating conditions’, (three items; individual’s perception of the availability of resources to use the technology; e.g., “I felt I had the necessary knowledge to use the remote monitoring equipment”), ‘effort expectancy’ (four items; ease-of-use related to using the technology; e.g., “I find it easy to use such equipment”), ‘performance expectancy’(four items; effectiveness to users in performing specific tasks; e.g., “I feel remote monitoring equipment is useful in obtaining health information”), and ‘task-technology fit’ (three items; the degree to which users believe that the performance of the technology match its intended use; e.g., “In general, the remote monitoring equipment fully met my needs”). All items will be rated using a five-point Likert scale (‘strongly disagree’ = 1, ‘strongly agree’ = 5), with higher scores reflecting greater scores towards its respective factor. Previous psychometric research has supported the validity and reliability of these items.[33]”

b. Patient experience of remote monitoring equipment sub-section

The TAM and the TBP were mentioned in the introduction, nevertheless is not clear whether questions regarding user acceptance were formulated based on these theoretical models?. Which model will be used in the analysis? It is a combination?

Should this section be named user acceptance?

Response: The items were formulated using an integrated model of the UTAUT and TTF models of user acceptance. This information is now included in the Methods section. Additionally, we have changed the name of the section to “user acceptance of remote monitoring equipment”.

Methods (pg 11):

“This multidimensional measure is comprised of items relating to the different factors of user technology acceptance, based on a recent integrated model consisting of factors from the UTAUT and TTF models (see [33]).”

We would like to sincerely thank you for taking the time to review our manuscript and for providing such valuable feedback that allows us to improve the quality of our work.

VERSION 2 – REVIEW

REVIEWER	Silva, Cícera Federal University of Campina Grande Centre of Teacher Education
REVIEW RETURNED	27-Jun-2021
GENERAL COMMENTS	Most of the issues raised by the reviewer have been supplemented or corrected.